# A national survey of antibacterial consumption in Sri Lanka

**Shalini Sri Ranganathan**[1]*, **Chandanie Wanigatunge**[2], **G. P. S. G. Senadheera**[3], **B. V. S. H. Beneragama**[4]

**1** Department of Pharmacology, Faculty of Medicine, University of Colombo, Colombo, Sri Lanka,
**2** Department of Pharmacology, Faculty of Medical Sciences, University of Sri Jayewardenepura, Sri Jayewardenepura, Sri Lanka, **3** Department of Pharmacy and Pharmaceutical Sciences, Faculty of Allied Health Sciences, University of Sri Jayewardenepura, Sri Jayewardenepura, Sri Lanka, **4** National focal point for combating antimicrobial resistance in Sri Lanka, Ministry of Health, Colombo, Sri Lanka

* sshalini@pharm.cmb.ac.lk, sshalini14@hotmail.com

## Abstract

### Introduction

Optimizing the use of antibacterial medicines is an accepted strategy to combat the antibacterial resistance. Availability of reliable antibacterial consumption (ABC) data is a prerequisite to implement this strategy.

### Objectives

To quantify and describe the national ABC in Sri Lanka and to examine any differences in the consumption between public and private sector.

### Methods

The methodology for this survey was adapted from World Health Organization (WHO) methodology for a global programme on surveillance of antimicrobial consumption. Aggregate data on national consumption of systemic antibacterials (J01- Anatomical Therapeutic Chemical Classification (ATC) for 2017 were retrospectively extracted from all available data sources and classified using ATC classification. Quantity of consumption was converted to Defined Daily Doses (DDDs). Data are presented as total consumption and comparison between the public and private sector. Selected key quality indicators of ABC were compared between these two sectors.

### Findings

From the available data sources, the total ABC in 2017 was 343.46 million DDDs. Private sector consumption accounted for 246.76 million DDDs compared to 97.96 million DDDs distributed to entire public sector by the Ministry of Health. Beta-lactam-penicillins antibacterial group accounted for 58.79% in public sector compared to 27.48% in private sector while macrolides, quinolones and other beta-lactam antibacterials accounted for 60.51% in the private compared to 28.41% in public sector. Consumption of Reserve group antibacterials

**Data Availability Statement:** Data cannot be shared publicly because they belong to many third parties. The data underlying the results presented in the study are available from the Medical Supplies Division, Ministry of Health (https://www.msd.gov.

lk), Sri Lanka, State Pharmaceutical Corporation (https://www.spc.lk), Sri Lanka and State Pharmaceutical Manufacturing Corporation (https://www.spmclanka.lk) and private companies. Data were given to us under the agreement that the raw data will not be shared. Others can access each dataset used in this study by approaching the respective institutions with letters from a recognized local ethics review committee and administrative authorities. If the required approval is obtained from the respective institutions, we confirm that others would be able to access each dataset in the same manner as the authors. We confirm that authors did not have any special access privileges to each dataset that others would not have.

**Funding:** The author(s) received no specific funding for this work.

**Competing interests:** The authors have declared that no competing interests exist.

was negligible, and limited to private sector. Watch category antibacterials accounted for 46%, 24% and 54% of the total, public and private sector consumption, respectively.

## Conclusions

A disproportionately higher use of broad spectrum and Watch category antibacterials was observed in the private sector which needs further study. This national consumption survey highlights the need and provides the opening for establishment of ABC surveillance in Sri Lanka.

## Introduction

Antimicrobial resistance (AMR) is a major public health challenge [1]. A global survey conducted by the World Health Organization (WHO) in 2014 has shown a high level of resistance to both first line and Reserve category antibacterials for nine pathogenic bacteria responsible for common infections in all WHO regions [2]. Infection with such resistant microorganisms result in longer illnesses, increased mortality, prolonged hospital stays and increased overall costs [1, 3, 4]. Antimicrobial resistance affects all areas of health including veterinary and environmental practices and impacts the entire society and its economy [2, 5].

Principally driven by low-and-middle-income countries (LMICs) the global antibacterial consumption (ABC), expressed in defined daily doses (DDD), has increased by 65% from 2000 to 2015 [6]. An increased use of broad-spectrum and last-resort antibacterials was observed in both LMICs and high-income countries (HICs) during this period [6]. If the present policies continue, ABC is expected to increase by 200% in 2030 compared to the consumption in 2015 [6].

The association between ABC and development of antibacterial resistance (ABR) is well documented and a reduction of inappropriate use of antibacterials could reduce development of resistance [7, 8]. Countering ABR needs long term strategies which include strengthening the healthcare systems and enacting regulations to ensure appropriate use of and access to antibacterial agents [5]. To enhance access to antibacterials for treatment of commonly occurring infections and their appropriate use, the WHO introduced the Access, Watch, and Reserve (AWaRe) classification of antibacterials as part of Essential Medicines List [9]. With the application of AWaRe classification, the WHO national level target is that 60% of the antibacterials used should be from the Access category by 2023 [9].

The WHO's Global Action Plan (GAP) for Antimicrobial Resistance (AMR) [5] calls for member states to put in place national plans to urgently combat AMR. Five strategic objectives have been identified to achieve the goals of the GAP [5]. The fourth objective viz to "Optimize the use of antimicrobial medicines in human and animal health" needs reliable antimicrobial consumption (AMC) data [5, 10]. Data on AMC are vital to understand AMR, as selection pressure due to use of antimicrobials is a preventable driver for development and spread of AMR [7, 8]. While data on AMC are collected and analysed in many high- and middle-income countries, there is limited data on AMC from lower-income countries [10]. However, the available data from LMICs show a greater increase in the use of Watch category antibacterials and a greater reduction in the Access to Watch ratio [6, 11] To effectively curtail AMR, surveillance data from AMR must be linked to that of AMC [12].

The WHO methodology for a global programme on surveillance of antimicrobial consumption provides a practical framework to obtain such data in resource limited countries (RLC)

[13]. This involves the collection of "Consumption" and "Use" data and recommends that countries separate "consumption data" from "use data" as the objectives, methods and outcomes for these two categories of data are different. "Consumption data" refers to estimates derived from of aggregated data, mainly derived from import, sales or reimbursement databases whereas "use data" refers to estimates derived from patient-level [13].

Sri Lanka is a lower middle-income country [14]. Both the public and the private sectors provide allopathic healthcare services in Sri Lanka but the share of care is different for inpatients and outpatients. The public sector provides the bulk of inpatient care while outpatient care is shared between both public and private sectors [15]. Infections are a leading cause of morbidity and mortality in the public sector health care institutions with zoonotic and other bacterial infections being the 2nd leading cause of death in 2017 with the highest case fatality rates seen from septicaemia and pneumonia [15]. A similar picture was seen in children where pneumonia and other bacterial infections were the 4th and 5th leading causes of death [15]. The absence of morbidity and mortality data from the private sector makes comparisons between the sectors difficult.

The country imports the bulk of its antimicrobials through the State Pharamceuticals Corporation (SPC), which is the State's procurement arm, and independent private importers. Limited amounts of antimicrobials are manufactured by the State Pharmaceutical Manufacturing Corporation (SPMC) and individual local manufacturers. The SPC is the sole supplier of antimicrobials to the public sector. It directly imports and also procures from local manufaturers and are distributed to medical institutions in the public sector by the Medical Supplies Division (MSD) of the Ministry of Health. When antibacterials are not available at the MSD, the individual hospitals have the option to procure them from retail pharmacies as "local purchases". Antibacterials for the healthcare institutions in the private sector are purchased from SPC, independent private importers and local manufacturers.

Sri Lanka has an established and successful AMR surveillance programme, coordinated by the Sri Lanka College of Microbiologists, but there is no system in place to obtain aggregated AMC data. The available AMR data shows significant resistance by bacteria causing common infections to 1st line antibacterials [16–18]. Available AMC data from Sri Lanka are either limited to pharmaceutical sales data which lacks information of the public sector ABC [6, 19] or only from the public sector and lacks information about consumption in the private sector [20]. The public sector data showed an increase of ABC by 143% (44.4–108.2 million DDDs) with a significant shift towards the use of broad-spectrum antibacterials from 1998 to 2018 [20]. There is, however, no system at present to correlate AMC/ABC data with AMR/ABR patterns in the country.

The Sri Lanka Association of Clinical Pharmacology and Therapeutics (SLACPT), in collaboration the National Focal Point for combating AMR in Sri Lanka, therefore conducted this national survey of antibacterial consumption for 2017. Although the WHO methodology [13] has defined a core set of antimicrobials namely antibacterials, antibacterials for alimentary tract and nitroimidazole derivatives for protozoal diseases that all countries should monitor in their surveillance programme, this study has surveyed only the antibacterial consumption (ABC).

Our objective was to quantify and describe the national antibacterial consumption in Sri Lanka and compare the consumption between public and private healthcare sectors. This paper presents the methods adopted, discusses the key findings and the problems encountered when conducting the survey. Our findings would be helpful when planning for comprehensive national ABC/AMC surveys or surveillances.

## Materials and methods

### Study design and data sources

The methodology of this study was adapted from WHO methodology for a global programme on surveillance of antimicrobial consumption [13]. It was a descriptive cross-sectional study in which aggregate data on antibacterial consumption in 2017 were retrospectively extracted from all available data sources in 2018. The WHO methodology recommends to survey anti-microbials including anti-protozoals, anti-fungals, anti-malarials and anti-virals in addition to antibacterial agents (ABAs). However, to start with, we have surveyed only the ABAs listed under antibacterials for systemic use (J01) in the Anatomical Therapeutic Chemical (ATC) classification system [21].

Local manufacturing of antibacterials is limited in Sri Lanka and the importers are the major supplier of antibacterials for the country. State Pharmaceuticals Corporation is the sole importer for public sector and Rajya Osusala Pharmacies (retail pharmacy chain of SPC). It also imports for private market. In addition, there are many importers who cater for private market. Considering the supply system in Sri Lanka, we approached the Sri Lanka Customs, Department of Imports and Exports, SPC, State Pharmaceutical Manufacturing Corporation (SPMC), Medical Supplies Division (MSD), private importers and the private manufacturers (list was obtained from the NMRA website, www.nmra.gov.lk accessed on 31st August 2018) for ABC data. As these data sources are expected to provide the data for the entire country, no sampling was done. Details of all antibacterials were requested irrespective of their essential medicine list or Sri Lankan Formulary listing status. We provided the ATC code (J01) for those who had the data in the ATC format and a list containing the names of antibacterials cat-egorized under J01 for those who did not have the data in the ATC format. A custom-made MS Excel worksheet was developed based on WHO methodology and our previous experience in ABC surveillance in Colombo district [22]. Data from the MSD were electronically trans-ferred. All other data sources submitted data in paper format and these were manually entered into the Excel worksheet. All the precautions were taken to ensure the accuracy of data entry. The WHO methodology recommends a detailed product-level electronic data to be collected for ABC surveillance programmes. However, for this survey, data which had the minimum details, name, dosage form, strength and quantity were considered as "complete" and included for analysis.

### Measures of antibacterial consumption

The WHO defines consumption data as "estimates derived from aggregated data sources such as import or wholesaler data, or aggregated health insurance data where there is no informa-tion available on the patients who are receiving the medicines or why they are being used" [13]. National antibacterial consumption (ABC) data provide a proxy estimate of use of these agents in the country. Antibacterials for systemic use (J01, ATC classification level 2) obtained from the data sources were further classified to level 3, 4 and 5. We have presented the con-sumption data at level 3 (therapeutic or pharmacological sub-group) and level 5 (chemical sub-stance). Quantity of consumption is expressed as Defined Daily Doses (DDDs) using the formula given below. The DDD is defined as "the assumed average maintenance dose per day for a medicine used for its main indication in adults". Total grams consumed was determined by summing the amounts of active ingredient across the various formulations (different strengths of tablets or capsules, syrup formulations) and pack sizes. The DDD value is assigned by the WHO Collaborating Centre and obtained from their website (http://www.whocc.no/atc_ddd_index/). Number of DDDs consumed was calculated by dividing the total

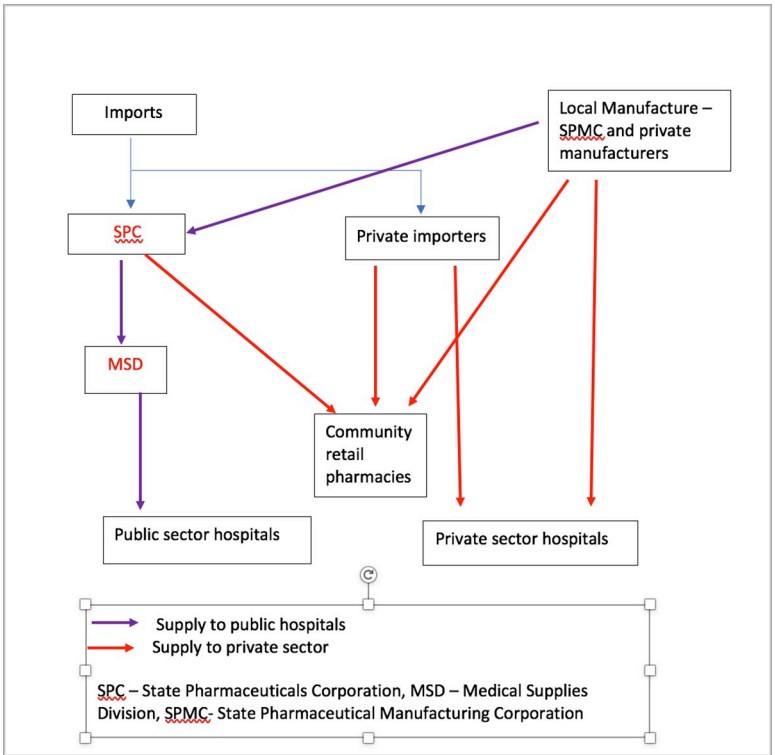

**Fig 1. Antibacterial supply chain in Sri Lanka.**

consumption (in grams) by DDDs (in grams). Though the WHO methodology recommends that the variables for consumption estimates should include packages and DDDs, we have used only DDDs as package details were not available for this retrospective survey. In addition to presenting the total consumption data obtained from different data sources, we have also compared the public sector data with that of private sector, keeping in mind that the private sector data could be an underestimate as it was impossible to verify the accuracy and completeness of data. The supply chain of antibacterials for public and private sectors in Sri Lanka is shown in Fig 1 which shows that the chances of duplication of data is minimal. We also have compared few key ESAC based quality indicators [23] of ABC as well as the volumes of antibacterials in the "Access, Watch and Reserve" categories (AWaRe classification) between these two sectors [9].

## Ethical considerations

Ethics Review Committee of Sri Lanka Medical Association exempted this survey from review (ERC/18/14). A formal letter of request with a copy of ethics review committee's letter of exemption was sent to all the institutions who had the data. Investigators personally visited many of these institutions to explain the aim of survey. Data presented here are from the consenting institutions. Data comprised aggregate data of the amount of antibacterials distributed, imported or manufactured by these institutions and not individual patient or hospital data.

## Results and discussion

We analyzed the data from the MSD, SPC, one local manufacturer (SPMC) and 12 private importers. Four local manufacturers did not provide data. Although 78 are registered as

private importers with the NMRA, the exact number of companies that actually imported antibacterials in 2017 was not available. The rough estimate from the NMRA records was 40 and 12 of them provided analyzable data. Four out of the 5 leading importers provided analyzable data whilst data provided by one major importer had to be discarded as the Company did not provide the information on strengths of the dosage forms which is essential to calculate DDD. The MSD data is reliable and is the almost complete data for public sector. The SPC provided data separately for public and private sector while SPMC provided cumulative data for both sectors. Data provided by the private importers is their imports for private sector. The SPC data for private sector may include the data from companies that have not provided their imports data to us (Fig 1).

The total volume of antibacterial agents (ABA) (in million DDDs) imported/distributed in 2017 by the respective agencies is shown in Table 1.

In Sri Lanka, most of the inpatient care is provided by the public sector while the outpatient care is shared between the public and private sectors [15]. Despite bulk of patient care being provided by the public sector, the ABC in the private sector (Table 1; data sources 1, 4 and 7) was 246.76 million DDDs compared to 97.96 million DDDs distributed to the public sector by the MSD, which is the sole supplier of medicines to the public sector (data source 2).

The SPC was initially established as a procurement agency for the public sector. However, in 2017 a significant 74.5% of its antibacterial imports were to the private sector (163 million DDDs,) while only 14.35% was to the public sector (31.43 million DDDs). The amount distributed to the public sector by MSD includes leftover stocks from 2016 and hence more than what it had received from the SPC in 2017.

The higher ABC of private sector could be due to the significant proportion of outpatient care that is provided by the private sector which includes the private hospitals and independent family physicians in the community. The antibacterials prescribed by these sectors are obtained from the retail pharmacies in the private hospitals and in the community and will contribute towards the increased ABC of the private sector.

## Comparison of ABC between public and private sectors

As Sri Lanka has a free public health care system and a fee levying private health care system, we analyzed the data according to the total ABC and in the two sectors. The top four groups of antibacterials consumed in Sri Lanka were beta-lactams, penicillins, other beta-lactams, macrolides and quinolones (Table 2). However, major differences were observed in the proportion of volumes consumed with these categories between the public and private sectors. The beta-lactam antibacterials, penicillins considerably outnumbered (58.57%) the other three groups (13.7%, 7.8%, and 6.8%) in the public sector whereas each of the top four groups

**Table 1. Total volume of ABAs (in million DDDs) imported/distributed in 2017 by the different agencies.**

| Agency | Volume of ABAs (in million DDDs) |
|---|---|
| 1. Distributed by SPC to private sector | 163.04 |
| 2. Distributed by MSD to public sector | 97.96 |
| 3. Distributed by SPMC to both to public and private sector | 61.18 |
| 4. Imported by private sector | 59.30 |
| 5. Manufactured by SPMC | 56.08 |
| 6. Distributed by SPC to MSD | 31.43 |
| 7. Distributed by SPC to its retail pharmacies (Rajya Osusala) | 24.41 |

**Table 2. Consumption of different pharmacological sub-groups of antibacterials in the country and public and private sectors.**

| ATC code | ATC level 3 classification of ABAs | ABA consumption volume in DDD per million | | | | | | DDD per 1000s population |
|---|---|---|---|---|---|---|---|---|
| | | Public sector (%) | | Private sector (%) | | Total (%) | | |
| J01A | Tetracycline's | 5.94 | (6.06) | 14.89 | 0.01 | 20.83 | (6.04) | 0.98 |
| J01B | Amphenicols | 0.00 | (0.00) | 0.19 | 5.90 | 0.19 | (0.06) | 0.01 |
| J01C | Beta-lactam antibacterials, penicillins | 57.38 | (58.57) | 67.80 | 2.62 | 125.18 | (36.31) | 5.90 |
| J01D | Other Beta-lactam antibacterials | 13.38 | (13.66) | 42.20 | 0.10 | 55.58 | (16.12) | 2.62 |
| J01E | Sulfonamides and trimethoprim | 0.50 | (0.51) | 1.71 | 3.22 | 2.21 | (0.64) | 0.10 |
| J01F | Macrolide, lincosamide and streptogramins | 7.69 | (7.78) | 60.57 | 0.01 | 68.26 | (19.80) | 3.22 |
| J01G | Aminoglycoside antibacterials | 0.14 | (0.14) | 0.00 | 2.51 | 0.14 | (0.04) | 0.01 |
| J01M | Quinolone antibacterials | 6.66 | (6.82) | 46.57 | 0.90 | 53.23 | (15.44) | 2.51 |
| J01X | Other antibacterials | 6.28 | (6.41) | 12.84 | 16.26 | 19.12 | (5.55) | 0.90 |
| | **Total** | **97.96** | **100.00** | **246.76** | **0.98** | **344.72** | **100.00** | **16.26** |

accounted for between 15–36% of the ABC in the private sector. Compared to the public sector, the consumption of macrolides, quinolones and other beta lactam antibacterials is disproportionately higher in the private sector.

The consumption of most frequently consumed antibacterials (5th and last level of ATC classification) within these top four groups are shown in Table 3. (S1 Table). Substantial differences were observed between public and private sectors in the consumption of individual antibacterials within each of the top four groups. In the private sector co-amoxiclav was the most consumed antibacterial in J01C group and azithromycin in the J01F group while amoxicillin and erythromycin were the equivalents in the public sector. Interestingly, benzyl penicillin was consumed only in the public sector.

**Table 3. Most consumed antibacterials of the top four pharmacological groups in private and public sectors.**

| ATC level 5 classification of ABAs (Code) under each level 3 classification | Name of classes and individual ABAs | Public sector (%) | | Private sector (%) | |
|---|---|---|---|---|---|
| | | DDDs | (%) | DDDs | (%) |
| J01C | Beta-lactam antibacterials, Penicillins | | | | |
| J01CA04 | Amoxicillin | 21.91 | (22.37) | 22.66 | (9.18) |
| J01CE01 | Benzyl penicillin | 16.95 | (17.30) | - | - |
| J01CF02 | Cloxacillin | 7.96 | (8.13) | 2.49 | (1.01) |
| J01CF05 | Flucloxacillin | 0.32 | (0.33) | 1.93 | (0.78) |
| J01CR02 | Co-Amoxiclav | 8.72 | (8.90) | 39.07 | (15.83) |
| J01D | Other Beta-lactam antibacterials | | | | |
| J01DB01 | Cephalexin | 4.57 | (4.67) | 16.12 | (6.53) |
| J01DC02 | Cefuroxime | 7.79 | (7.95) | 20.91 | (8.47) |
| J01DD08 | Cefixime | 0.05 | (0.05) | 4.59 | (1.86) |
| J01F | Macrolide, Lincosamide and Streptogramins | | | | |
| J01FA01 | Erythromycin Stearate | 3.52 | (3.59 | 6.13 | (2.49) |
| J01FA09 | Clarithromycin | 2.67 | (2.73) | 13.58 | (5.50) |
| J01FA10 | Azithromycin | 1.26 | (1.29) | 38.74 | (15.70) |
| J01M | Quinolone antibacterials | | | | |
| J01MA01 | Ofloxacin | 0.03 | (0.03) | 0.32 | (0.13) |
| J01MA02 | Ciprofloxacin | 5.81 | (5.93) | 34.93 | (14.16) |
| J01MA06 | Norfloxacin | 0.52 | (0.53) | 2.76 | (1.12) |
| J01MA12 | Levofloxacin | 0.22 | (0.22) | 8.05 | (3.26) |

Quality indicators of ABC for the country as a whole and separately for public and private sectors are given in Table 4. Indicators are calculated for total data to show how one sector affects the country data.

## Antibacterials consumption according to WHO AWaRe Classification

Fig 2 shows the consumption of antibacterials in the "Access, Watch and Reserve" categories for the country as a whole and separately for public and private sector (S2 Table).

Of the total antibacterials consumed, 54.19% were from the Access category while 45.57% were from the Watch group with an Access: Watch ratio of 1.18. However, in the public sector this ratio was 3.16 while it was 0.84 in the private sector.

The single most important difference observed between public and private sector (Tables 2–4, Fig 1) was disproportionately higher use of broad spectrum antibacterials in the private sector. We compared the top ten oral and parental antibacterials used between the sectors (Tables 5 and 6).

While both sectors have consumed large amounts of oral and parenteral antibacterials in the Watch category, this is much more in the private sector. Notable is the disparity in the consumption of ceftriaxone, cefuroxime and meropenem. This is a concern as it would contribute significantly towards the spread of antibacterial resistant bacteria in the country, Surveillance data on both resistance and consumption are essential to obtain a comprehensive picture of antibacterial resistance. Correlating ABC data with the ABR patterns will help to identify areas that need further action. While national data on ABR patterns are available in Sri Lanka [24, 25], to the best of our knowledge this is the first attempt at obtaining national antibacterial consumption data. Previous data from Sri Lanka which have been included in other surveys [6, 19, 20] have been obtained from either only pharmaceutical sales [6, 19] data or only public sector ABC data [20]. Adopting WHOs' standard methodology made it possible for us to compare our findings with similar studies done globally and in the region.

We compared Sri Lanka's ABC in 2017 with global surveys on ABC [6, 11] and that of the ESAC-Net countries [26]. In 2015, the most commonly consumed antibacterial classes globally were broad-spectrum penicillins (J01CA), cephalosporins (J01D), quinolones (J01M) and macrolides (J01F) [6, 11]. This was similar to the top four groups of antibacterials consumed in Sri Lanka. The average total ABC for systemic use (ATC group J01) in the EU/EEA (23.4

**Table 4. Comparison of few key quality indicators of ABC between public and private sector.**

| | Indicator | Public sector | Private sector | Total |
|---|---|---|---|---|
| J01CE_% | Consumption of β-lactamase sensitive penicillins (J01CE) expressed as percentage of the total consumption of antibacterials for systemic use (J01) | 18.77 | 0.18 | 5.46 |
| J01CR_% | Consumption of combination of penicillins, including β -lactamase inhibitor (J01CR) expressed as percentage of the total consumption of antibacterials for systemic use (J01) | 8.98 | 15.83 | 13.89 |
| J01DD +DE_% | Consumption of third and fourth generation of cephalosporins (J01(DD+DE)) expressed as percentage of the total consumption of antibacterials for systemic use (J01) | | | |
| | 1st Generation | 4.67 | 6.55 | 6.01 |
| | 2nd Generation | 7.95 | 8.52 | 8.35 |
| | 3rd Generation | 0.72 | 1.98 | 1.62 |
| | 4th Generation | 0.00 | 0.00 | 0.00 |
| J01MA_% | Consumption of flouroquinolones (J01MA) expressed as percentage of the total consumption of antibacterials for systemic use (J01) | 6.71 | 18.81 | 15.31 |
| J01_B/N | Ratio of the consumption of broad (J01(CR+DC+DD+(F-FA01))) to the consumption of narrow spectrum penicillins, cephalosporins and macrolides (J01(CE+DB+FA01)) | 0.81 | 5.25 | 2.86 |

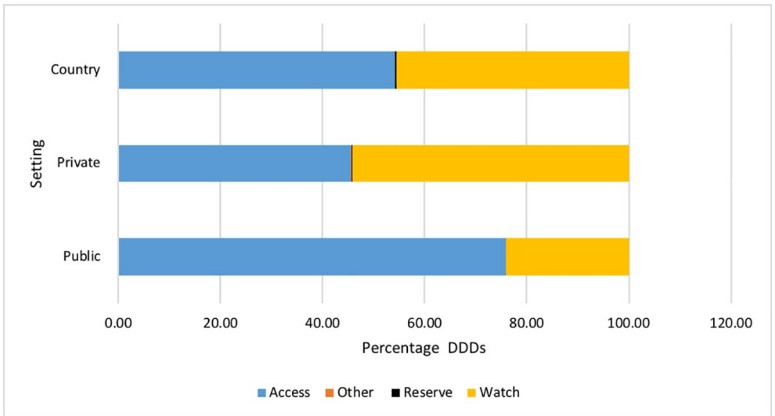

**Fig 2. Consumption of antibacterials in Access, Watch and Reserve categories.**

DID) [26] was much higher than that of Sri Lanka for 2017 (16.26 DID). A key finding was the ratio of the consumption of broad spectrum to narrow spectrum penicillins, cephalosporins and macrolides (J01_B/N). This was 2.86 for the country, 0.81 for the public sector but a significant 5.25 for the private sector (Table 4). This ratio is higher than what was seen in the community consumption for ESAC-Net countries i.e. 2.25 [26]. The higher DID and the comparatively lower J01-B/N in ESAC-Net countries could be due to greater access to and better regulation of antimicrobials in these countries.

Direct comparison with other WHO South East Asian Region (SEAR) countries in the region was limited by the lack of studies and the disparity of data sources in different studies. Although South East Asian Region (SEAR) was excluded in the WHO report on surveillance of antibacterial consumption from 2016–2018 due to the lack of data, a high level of consumption of cephalosporins and quinolones was found in some countries of the region [2]. A higher consumption of other beta-lactam antibacterials, macrolides and quinolones was seen in the private sector compared to that of the public sector of Sri Lanka. This was somewhat similar to that of the retail private sector in India which also showed an increase in the use of 3<sup>rd</sup> generation cephalosporins, penicillins with beta-lactamase inhibitors, and of newer classes of antibacterials like carbapenems, lincosamides and glycopeptides [27].

As the WHO recommends that the Watch group should be prioritized as key targets of stewardship programs and monitoring as they have higher resistance potential [9], we analysed

**Table 5. Comparison of the top ten oral antibacterials consumed between sectors.**

| Public sector | | | | Private Sector | | | |
|---|---|---|---|---|---|---|---|
| ABM | AWaRe group | DDDs in million | % | ABM | AWaRe group | DDDs in million | % |
| Amoxicillin | A | 21.91 | 28.96 | Co-Amoxiclav | A | 38.95 | 15.89 |
| Cloxacillin | A | 7.82 | 10.33 | Azithromycin | W | 38.74 | 15.81 |
| Cefuroxime | W | 6.7 | 8.86 | Ciprofloxacin | W | 34.91 | 14.25 |
| Co-Amoxiclav | A | 6.2 | 8.2 | Amoxicillin | A | 22.66 | 9.25 |
| Doxycycline | A | 5.94 | 7.85 | Cefuroxime | W | 20.7 | 8.45 |
| Ciprofloxacin | W | 5.81 | 7.68 | Cephalexin | A | 16.12 | 6.58 |
| Metronidazole | A | 4.62 | 6.11 | Doxycycline | A | 14.35 | 5.86 |
| Cephalexin | A | 4.57 | 6.04 | Clarithromycin | W | 13.49 | 5.5 |
| Erythromycin | W | 3.52 | 4.66 | Metronidazole | A | 12.03 | 4.91 |
| Clarithromycin | W | 2.62 | 3.47 | Levofloxacin | W | 8.03 | 3.28 |

**Table 6. Comparison of the top ten parental antibacterials consumed between sectors.**

| Public sector | | | | Private sector | | | |
|---|---|---|---|---|---|---|---|
| ABM | AWaRe group | DDDs in million | % | ABM | AWaRe group | DDDs in million | % |
| Benzyl penicillin | A | 16.95 | 75.98 | Flucloxacillin | A | 0.35 | 20.57 |
| Co-amoxiclav | A | 2.52 | 11.29 | Ampicillin | A | 0.24 | 14.43 |
| Cefuroxime | W | 1.08 | 4.86 | Ceftriaxone | W | 0.23 | 13.53 |
| Ceftriaxone | W | 0.48 | 2.16 | Cefuroxime | W | 0.21 | 12.39 |
| Meropenem | W | 0.3 | 1.35 | Metronidazole | A | 0.14 | 8.53 |
| Cefotaxime | W | 0.14 | 0.64 | Meropenem | W | 0.12 | 7.26 |
| Cloxacillin | A | 0.14 | 0.62 | Co-amoxiclav | A | 0.12 | 6.92 |
| Flucloxacillin | A | 0.13 | 0.56 | Clarithromycin | W | 0.09 | 5.32 |
| Gentamicin | A | 0.11 | 0.48 | Moxifloxacin | W | 0.04 | 2.27 |
| Levofloxacin | W | 0.09 | 0.38 | Cefotaxime | W | 0.03 | 1.96 |

the data according to WHO's AWaRe classification (9). Despite a 54.19% overall use of Watch antibacterials in the country, a higher consumption of these antibacterials was seen in the private sector (54.11%) compared to the public sector (24.11%) (Fig 2).This may be an under representation of the consumption of Watch antibacterials in both sectors as local purchase by public sector hospitals and direct purchase from importers by private sector hospitals and retail pharmacies have not been completely captured in our data especially as we had to discard data from a major importer. The disparity could be still higher as we do not have complete data from the private sector imports. A similar pattern where more antibacterials in the Watch group are consumed is also seen in several European countries and Japan [10] and in India [16, 28]. The higher consumption of Watch group antibacterials in the private sector in Sri Lanka is a concern especially as there is no data to suggest that the causative bacteria and/or their antibacterial sensitivity patterns differ between the sectors. This is more alarming as there is an increasing emergence of multi drug resistant bacteria in the country [16–18, 29]. It has also been shown that some of the pathogens causing lower respiratory tract infections are resistant to the first line Access antibacterials but show an increased sensitivity to 2nd line (Watch) agents [30]. Infections with these resistant bacteria to first line antibacterials would add a significant burden to the health budget.

A significant finding from our survey is the disproportionately higher use of broad-spectrum antibacterials in the private sector when compared to the ABC of the public sector. The disparity is difficult to explain, especially as prescribers in both these sectors being largely the same. In addition, the National Antibacterial Guidelines for empirical treatment of infections [31] have been widely disseminated to the prescribers and they are expected to adhere to these irrespective of the place of practice. A major factor that affects antibacterial consumption in the public hospitals in Sri Lanka is the highly limited hospital formulary available to prescribers. The formulary is based on the country's EMLs and the medicines are made available strictly according to it to the public sector institutions. Even though prescribers have the facility to "local purchase" medicines, this does not give them the same availability of medicines as those practicing in the private sector as the patients who seek care from these institutions are largely from lower income segments of the society. The freedom to prescribe any medicine that is available in the country and improved financial status of the patients who seek treatment from the private sector could contribute to the greater use of broad-spectrum and newer antibacterials in the private sector.

However, inappropriate prescriptions of antibacterials are seen in the outpatient management of respiratory infections in the public sector hospitals too [32, 33]. Patients demand for

antibacterials as a "quick fix" for infections, incorrect physician perception of the need for antibacterials, fear of bacterial super infection in viral diseases and the high patient volume seen in outpatient settings limiting the time for assessment have led to a greater prescription of broad spectrum antibacterials [33].

The Government of Sri Lanka has initiated many regulatory mechanisms to curtail inappropriate antibacterial use. All antibacterials are registered under Schedule 11B by the National Medicines Regulatory Authority and are prescription only [34]. Despite such regulations it is still possible to obtain antibacterials from retail pharmacies without a valid prescription [35, 36]. Self-medication of antibacterials by patients [36] and the freedom to access any practitioner without an appropriate referral which could lead to duplication and/or inappropriate prescriptions, also contribute to an increase in ABC and AMR in the country.

While we do not have complete national data, this paper presents the maximum possible extractable data on ABC in Sri Lanka for the year 2017. As recommended by the WHO for counties which are starting antimicrobial surveillance, we have used procurement/issues data available at the central. This, however, does not reflect what is actually consumed by the end user. The key strength of our study is usage of standardized WHO methodology for reporting ABC in DDD and using ATC classification. The DDD methodology allowed us to use aggregated antibacterial purchase data and made it possible to compare our data with regional and global data. However, the DDD may not reflect the prescribed daily dose (PDD) for individual patients and cannot be used to measure consumption in paediatric wards since the measure is based on adult dosing [37]. It also does not accurately measure antibacterial consumption in cases of renal or hepatic dysfunction, often underestimating the actual antibacterial usage [37].

An important limitation of our study was the inability to capture all national data. This was largely due to incomplete and inadequate record keeping by the Customs and private importers. The inability of the SPMC to provide consumption data based on the sector to which it supplied added to the incomplete national consumption data. For meaningful interpretation of data, the total numbers of DDDs derived as consumption estimates should be adjusted for the population to which the data apply. Despite these limitations we have adjusted for the population (DID) to compare with similar studies as there is no separate DDD for children [21]. Therefore, we had to use the DDDs for adults in the calculations although both adults and children would have consumed the antibacterials.

This is the first time an attempt has been made to document the national consumption of antibacterials in Sri Lanka. While there are some limitations and the actual consumption could be an under estimation, we are confident that the pattern of antibacterial consumption documented in this paper is unlikely to change even if we have all the data on antibacterial consumption. The generalizability of our findings would depend on the systems in place to regulate and survey antibacterial consumption. The paper highlights the need for better regulation of antibacterial consumption and the need for robust surveillance systems. The latter could be both labour and resource intense to LMICs like Sri Lanka.

## Conclusions

Despite limitations our study provides the first national ABC data from Sri Lanka which captures the use in both public and private sectors. Although limited to the health care sector, it highlights the problems a LMIC face when trying to apply globally accepted survey methods to evaluate aspects of ABC. Establishing a central unit to coordinate all activities related to both AMR and AMC, using accepted classification when coding imports, getting the private sector into the programme and creating a central data base which records, analyses and generates statistics routinely are key areas that need attention for better surveillance of antimicrobial use.

As antibacterials use is to some extent patient and prescription driven in Sri Lanka, education campaigns targeting both patients and prescribers are needed to change behaviours and prescribing habits. Better implementation of existing regulation is vital to curtail antibacterial misuse.

## Recommendations for policy makers

We strongly recommend the establishment of robust and sustainable surveillance systems to periodically survey and monitor antibacterial consumption. A central body to coordinate the activities of antibacterial consumption is crucial. Surveillance systems should be developed, and adequate funding and resources to collect and analyze data should be made available. All data should be coded at the point of entry using the ATC classification which would help in analysis of data and to compare the consumption trends with other countries.

The data on ABC should be linked with that of ABR to identify trends of antibacterial use and changes in antibacterial sensitivity patterns. The ABC should be reviewed annually to identify trends of use and to regularize antibacterial consumption. Based on the surveillance data, national policies and guidelines for antibacterial use should be developed and measures should be in place to ensure that they are adhered to. Linking up with the WHO programme that has been introduced for LMICs [38] is important to compare the country's activities with others.

## Supporting information

**S1 Table. Comparing the volume of antibacterial consumption (ATC classification Level 5) between private and public sector.**
(DOCX)

**S2 Table. AWaRe categorization of all the antibacterials.**
(DOCX)

## Acknowledgments

We acknowledge the Sri Lanka Association of Clinical Pharmacology and Therapeutics (SLACPT) for funding the data entry personnel, Mr P.A.A.S.P. Kumara for assisting in data entry, Dr Malitha Rubesinghe for coordinating approvals and all the officials and pharmacists who facilitated data extraction.

## Author Contributions

**Conceptualization:** Shalini Sri Ranganathan, Chandanie Wanigatunge.

**Formal analysis:** G. P. S. G. Senadheera.

**Methodology:** Shalini Sri Ranganathan.

**Project administration:** Shalini Sri Ranganathan, Chandanie Wanigatunge, G. P. S. G. Senadheera, B. V. S. H. Beneragama.

**Supervision:** Shalini Sri Ranganathan.

**Writing – original draft:** Shalini Sri Ranganathan, Chandanie Wanigatunge.

**Writing – review & editing:** Shalini Sri Ranganathan, Chandanie Wanigatunge, B. V. S. H. Beneragama.

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
