## [Decision Letter · Decision Letter 0]

17 May 2021

PONE-D-21-11873

National surveillance of antibacterial consumption in Sri Lanka

PLOS ONE

Dear,

Thank you for submitting your manuscript to PLOS ONE. After careful consideration, we feel that it has merit but does not fully meet PLOS ONE’s publication criteria as it currently stands. Therefore, we invite you to submit a revised version of the manuscript that addresses the points raised during the review process.

We look forward to receiving your revised manuscript.

Kind regards,

Muhammad Shahzad Aslam, Ph.D.,M.Phil., Pharm-D

Academic Editor

PLOS ONE

Journal Requirements:

Reviewers' comments:

Reviewer's Responses to Questions

**Comments to the Author**

1. Is the manuscript technically sound, and do the data support the conclusions?

Reviewer #1: Yes

Reviewer #2: Partly

Reviewer #3: Yes

2. Has the statistical analysis been performed appropriately and rigorously? 

Reviewer #1: N/A

Reviewer #2: No

Reviewer #3: Yes

3. Have the authors made all data underlying the findings in their manuscript fully available?

Reviewer #1: Yes

Reviewer #2: Yes

Reviewer #3: Yes

4. Is the manuscript presented in an intelligible fashion and written in standard English?

Reviewer #1: Yes

Reviewer #2: No

Reviewer #3: Yes

5. Review Comments to the Author

Reviewer #1: The manuscript addresses a timely topic which is very important for all the developing countries. It also stresses the possibility of antibiotic abuse (especially second line antibiotics) within the private sector which can be controlled with appropriate measures as discussed in the article. It is well written except for a few changes as per my comments. Statistically, they have only used percentages which is descriptive enough given the content of the article. Please delete the words that were highlighted in the Introduction and add the comment I stressed to the discussion.

Reviewer #2: The manuscript entitled “National surveillance of antibacterial consumption in Sri Lanka” is a descriptive study aimed to quantify and describe national antimicrobial consumption. It is very important concept and has practical policy implications; however, authors need to address the following major comments/suggestions to improve the quality of the paper before it will come to publication.

General comment

The manuscript needs to be reformatted by referring the journal style, excessive bolding of words and excessive unnecessary abbreviations should be reduced, and appropriate citations, repeated editing and proof reading are required. In addition to making the paper mind-numbing, those things are very technical and reduce the readability of the paper, and hence, it needs careful revisions.

Detail comments/sugessions

Title

• I suggest the title need to be revised. I think it is a cross sectional study, which is a survey, but the title saying surveillance. Unlike your study, surveillance is an ongoing collection of information to detect changes or it is repeated survey.

Abstract

• The abstract lacks background.

• The word WHO, in method section need to be written in full. You should first write the full term before you abbreviated. Full term do not need to be in caps and brackets, just abbreviations. Similarly for ABC in the finding section of abstract.

• The sentence “Reserve and watch category antibacterials accounted for 46, 24 and 54% of the total, public and private sector consumption” not clear and needs revisions.

• The aim of the study and the conclusion is mismatched. The aim is to describe and quantify, but the conclusion seems comparisons between public and private. The recommendation “Our study has provided the evidence that antibacterial surveillance is possible in resource limited countries and it must be made mandatory” needs revisions. Because, the recommendation is beyond the scope of the study, recommended to other resource limited countries are no appropriate, as the method including sampling procedure used in the study are not allowed to infer to other countries. I also suggest to not use the word mandatory since recommendation are not approached as obligatory.

• I not saw key words. Please add.

• The background, objective, method, results and conclusions of manuscript need to be reformatted including make them bold.

Background

• Revise the typo error (two point after the title) of “Introduction:”

• The paper generally needs serious citation revisions. For example, in the first paragraph only, from five sites that need citations, only one has citation and the rest four are not. Please, also add more evidences/citations for those cited also.

• The third paragraph of introduction also missed citations and needs revised again.

• The last paragraph of introduction “therefore conducted a national surveillance of antibacterial consumption (ABC)” needs revisions. As suggested above to the title, this study is not surveillance rather a survey. The two concepts are quite different. What are the significance of the sentence “This paper outlines the methods adopted, key findings and recommendations for establishment of a national surveillance programme”.

• The background section lacks and needs to revise based on updated prior literatures on global, regional and national picture about the antimicrobial consumption.

• Authors also need to add contribution of this study and practical policy implication.

• I couldn’t see the aim/s and or objective of the study in this section that is mentioned in the abstract, authors need to expand those points including the rationale of the study.

Methods

• There are many bolded disorganized subsections. No need to say background here. Please reformat them based on the journal style. For instance, study design, data source and sampling procedure in to one subsection. Variables of the study (outcome variable vs explanatory variables, including their definitions with appropriate citations, coding and categories in another subsection. Then, data entry, data analysis, and presentation of results as statistical analysis, the ethical issues in another subsection like that. Please elaborate a little more on the ethical issues. The ethical issues mentioned in the data source section need to be bring in to ethical consideration section.

• Please summarize data source and availability of data into one.

• Authors mentioned as it indicates the 2017 national antimicrobial consumption. However, the data collection period or this specific study is not mentioned.

• I couldn’t found about sampling technique and procedures use in the study. Who were the source and sampled population? How they were selected? Who were the data collectors? What are the tools?

Results

• The results need to be summarized with subsection to make clearer of the key findings.

• Please avoid the typo error (colon after the title of results).

Discussion

• The discussion is poor. It seems redundancy with results, and therefore, needs careful revisions; compare and contrast of your key findings with prior literatures and justify or discuss in details especially for inconsistency findings. It also have citations problem.

Conclusions

The conclusion is not specific and not aligned with the findings. Please first write the word in full before you abbreviated it such as LMICs. You recommend to LMICS, but I couldn’t see any evidence in the method sections including sampling technique, that support for generalizability of the findings for LMICs. The conclusion is beyond the study’s scope and not specific. Please revise it.

Reviewer #3: This is a comprehensive study looking at antibiotic consumption patterns in a LMIC. It is well-written except for a few grammatical errors (Introduction - line 3, page 5 - line 10, page 20 - lines 1 and 6). Data however is far from complete since many potential sources have not been able to provide usable data (ie only 12/78 private importers have submitted usable data).

A major factor that affects antibiotic consumption in the public hospitals in Sri Lanka is the highly limited formulary that is available for prescribers, compared to the private sector. This fact has not been taken in to account in the Discussion. Being able to 'locally purchase' antibiotics in the public sector does not equate to the choice of antibiotics available in the private sector.

No information on antibiotic resistance patterns in the country is provided. Therefore, the usefulness and applicability of the prescription patterns is not explored in detail.

6. PLOS authors have the option to publish the peer review history of their article (what does this mean?). If published, this will include your full peer review and any attached files.

Reviewer #1: **Yes: **Dr Chanika Alahakoon (MBBS, MPhil), Department of Physiology, Faculty of Medicine, University of Peradeniya, Sri Lanka

Reviewer #2: **Yes: **Betregiorgis Zegeye

Reviewer #3: No

---

## [Author Response · Author response to Decision Letter 0]

10 Jun 2021

PONE-D-21-11873

National surveillance of antibacterial consumption in Sri Lanka

Amended title: A National survey of antibacterial consumption in Sri Lanka

Response to Reviewers 

COMMENTS RESPONSE FROM AUTHORS

ACADEMIC EDITOR 

 Thank you very much

We apologize for not following the journal style 

We have now revised the whole manuscript based on the journal style guide for title page, main body, tables, figure and references 

Please provide additional details regarding participant consent. In the ethics statement in the Methods and online submission information, please ensure that you have specified (1) whether consent was informed and (2) what type you obtained (for instance, written or verbal, and if verbal, how it was documented and witnessed). If your study included minors, state whether you obtained consent from parents or guardians. If the need for consent was waived by the ethics committee, please include this information.

If you are reporting a retrospective study of medical records or archived samples, please ensure that you have discussed whether all data were fully anonymized before you accessed them and/or whether the IRB or ethics committee waived the requirement for informed consent. If patients provided informed written consent to have data from their medical records used in research, please include this information. Thank you 

This is a retrospective study of medicine supplies/imports data

Ethics Review Committee of Sri Lanka Medical Association exempted this survey from ethics review committee approval (ERC-18/14). Formal request letter with a copy of ethics review committee approval was sent to all the institutions who had the data. Investigators personally visited many of these institutions to explain the aim of survey. Data presented here are from the consenting institutions. Data comprised aggregate data of amount of antibacterials distributed, imported or manufactured by these institutions and not individual patient or hospital data. 

We have stated this under ethical considerations in the methods section

We note that you have indicated that data from this study are available upon request. PLOS only allows data to be available upon request if there are legal or ethical restrictions on sharing data publicly. For information on unacceptable data access restrictions, please see http://journals.plos.org/plosone/s/data-availability#loc-unacceptable-data-access-restrictions

Please see below (cover letter comment)

Cover letter

The data underlying this study belong to many third parties, Ministry of Health, State Pharmaceutical Corporation, State Pharmaceutical Manufacturing Corporation and private Pharmaceutical Companies. Raw data used to analyse antibiotic consumption in this manuscript is either state owned or private company/industry owned. Data were given to us under the agreement that the raw data will not be shared. Hence, we are unable to provide raw data in public domain. We confirm that we did not have any special access to the data that other researchers would not have.

REVIEWER 1 

The manuscript addresses a timely topic which is very important for all the developing countries. It also stresses the possibility of antibiotic abuse (especially second line antibiotics) within the private sector which can be controlled with appropriate measures as discussed in the article. It is well written except for a few changes as per my comments. Statistically, they have only used percentages which is descriptive enough given the content of the article. Please delete the words that were highlighted in the Introduction and add the comment I stressed to the discussion. Thank you very much. The typo has been deleted.

Add another limitation. The fact that you calculated the doses assuming only adults uses the antibiotics

 Thank you for this. We have added this limitation and appropriate reference has been cited. 

Grammatical and typo errors Thank you and have been corrected

REVIEWER 2 

General comments

The manuscript needs to be reformatted by referring the journal style, excessive bolding of words and excessive unnecessary abbreviations should be reduced, and appropriate citations, repeated editing and proof reading are required. In addition to making the paper mind-numbing, those things are very technical and reduce the readability of the paper, and hence, it needs careful revisions. Thank you very much for the comment 

We have carefully gone through the manuscript again and reformatted addressing the issues pointed out

We have revised the entiremanuscript and added the relevant citations

The background, objective, method, results and conclusions of manuscript need to be reformatted including make them bold. Thank you very much

We have reformatted the subheadings as per the Journal style 

Detail comments/suggestions

Title

I suggest the title need to be revised. I think it is a cross sectional study, which is a survey, but the title saying surveillance. Unlike your study, surveillance is an ongoing collection of information to detect changes or it is repeated survey. Thank you very much for the comment

We have amended the title as 

A National survey of antibacterial consumption in Sri Lanka

Abstract 

The abstract lacks background Thank you very much

We have added background in the revised manuscript 

The word WHO, in method section need to be written in full. You should first write the full term before you abbreviated. Full term do not need to be in caps and brackets, just abbreviations. Similarly for ABC in the finding section of abstract. Thank you very much

We have addressed this issue in the abstract as well as in the rest of the manuscript whenever applicable 

The sentence “Reserve and watch category antibacterials accounted for 46, 24 and 54% of the total, public and private sector consumption” not clear and needs revisions Thank you very much

We apologize for the error

We have amended in the revised manuscript 

Amended sentence is: 

Watch category antibacterials accounted for 46%, 24% and 54% of the total, public and private sector consumption respectively

The aim of the study and the conclusion is mismatched. The aim is to describe and quantify, but the conclusion seems comparisons between public and private. The recommendation “Our study has provided the evidence that antibacterial surveillance is possible in resource limited countries and it must be made mandatory” needs revisions. Because, the recommendation is beyond the scope of the study, recommended to other resource limited countries are no appropriate, as the method including sampling procedure used in the study are not allowed to infer to other countries. I also suggest to not use the word mandatory since recommendation are not approached as obligatory Thank you very much. We agree with the comment

We have amended the aim of the study as 

To quantify and describe the national antibacterial consumption in Sri Lanka and to examine any differences in the consumption between public and private sector

Conclusion is amended as 

A disproportionately higher use of broad spectrum and Watch category antibacterials was observed in the private sector which needs further investigation. This national annual consumption survey highlights the need and provides has provided the opening for establishment of an ABC surveillance in Sri Lanka

I not saw key words. Please add. We have given the key words in the online system

These were the key words: antibacterials; consumption; surveillance; antibacterial resistance; utilization (if possible, we will edit the surveillance as survey in the system)

Background

Revise the typo error (two point after the title) of “Introduction:” Thank you very much

We have corrected this typo error 

The paper generally needs serious citation revisions. For example, in the first paragraph only, from five sites that need citations, only one has citation and the rest four are not. Please, also add more evidences/citations for those cited also. Thank you.

We have revised the paper extensively and added citations as appropriate 

The third paragraph of introduction also missed citations and needs revised again. Thank you, this has been done. As we have redone the entire paper this is now paragraph 4.

The last paragraph of introduction “therefore conducted a national surveillance of antibacterial consumption (ABC)” needs revisions. Thank you, revised as given below

The Sri Lanka Association of Clinical Pharmacology and Therapeutics (SLACPT), in collaboration the National Focal Point for combating AMR in Sri Lanka, therefore conducted this national survey of antibacterial consumption (ABC) for 2017.

As suggested above to the title, this study is not surveillance rather a survey. The two concepts are quite different. Thank you very much

We have amended the term surveillance to survey 

What are the significance of the sentence “This paper outlines the methods adopted, key findings and recommendations for establishment of a national surveillance programme” This sentence has been modified as below:

This paper presents the methods adopted, discusses the key findings and the problems encountered when conducting the survey.

The background section lacks and needs to revise based on updated prior literatures on global, regional and national picture about the antimicrobial consumption. Thank you and the background section has been extensively revised with reference to other studies on global, regional and national AMC.

Authors also need to add contribution of this study and practical policy implication. Thank you, we have added the following:

Our findings would be helpful when planning for comprehensive national AMC surveys or surveillances.

I couldn’t see the aim/s and or objective of the study in this section that is mentioned in the abstract, authors need to expand those points including the rationale of the study. Thank you very much

We have included the objective of the study in the last paragraph of the introduction

Methods

There are many bolded disorganized subsections. No need to say background here. Please reformat them based on the journal style. For instance, study design, data source and sampling procedure in to one subsection. Variables of the study (outcome variable vs explanatory variables, including their definitions with appropriate citations, coding and categories in another subsection. Then, data entry, data analysis, and presentation of results as statistical analysis, the ethical issues in another subsection like that. Thank you very much for the comment

We have reformatted the methods section with three sub section 

Materials and Methods 

1. Study design and data source

2. Measures of antibacterial consumption

3. Ethical considerations 

Information given in the original manuscript had been reassigned to these three sub sections 

Please elaborate a little more on the ethical issues. The ethical issues mentioned in the data source section need to be bring in to ethical consideration section. Thank you very much for the comment

We have given some additional information under ethical considerations

Ethics Review Committee of Sri Lanka Medical Association exempted this survey from ethics review committee approval (ERC-18/14), Formal request letter with a copy of ethics review committee approval was sent to all the institutions who had the data. Investigators personally visited many of these institutions to explain the aim of survey. Data presented here are from the consenting institutions. Data comprised aggregate data of amount of antibacterials distributed, imported or manufactured by these institutions and not individual patient or hospital data. 

Please summarize data source and availability of data into one. Thank very much

Please see our responses for the first comment in the methods 

Authors mentioned as it indicates the 2017 national antimicrobial consumption. However, the data collection period or this specific study is not mentioned. We have already given this information 

Second sentence in the methods

It was done in 2018

I couldn’t found about sampling technique and procedures use in the study. Who were the source and sampled population? How they were selected? Based on the medicine supply system in Sri Lanka, the data sources we identified are expected to provide the data for the entire country. So, we did not employ any sampling technique

We have now given one statement to this effect in lines 25 and 26 under methods 

These data sources are expected to provide the data for the entire country. Hence, no sampling was done.

Who were the data collectors? What are the tools? Data comprised aggregate data 

They were extracted electronically (Excel worksheet) or manually transferred from the paper format of data submitted by the institutions

Hence we have not used data collectors. 

We have now given one statement to this effect in lines 30- 32 under methods 

Data from the MSD were electronically transferred. All other data sources submitted data in paper format which were manually entered in the Excel worksheet.

Methods- Additional change done by us 

In the methods section we have given a formula about calculating defined daily doses. To improve the format of the section, we have converted the formula into text 

Number of DDDs consumed was calculated by dividing the total consumption (in grams) by DDDs (in grams).

Results 

The results need to be summarized with subsection to make clearer of the key findings. Thank you. We have added subsections as appropriate

Please avoid the typo error (colon after the title of results Thank you and done

Results – Additional change/s done by us 

1. We have combined results and discussion under one title as per the journal format

2. Additional column in table 2 - DDD per 1000s population

3. Additional column in table 4 to show the quality indicators for the country

4. Changed figure 2 to tables 5 and 6 to highlight the issues clearly

Discussion 

The discussion is poor. It seems redundancy with results, and therefore, needs careful revisions; compare and contrast of your key findings with prior literatures and justify or discuss in details especially for inconsistency findings. It also have citations problem. Thank you. The discussion has been extensively revised to compare and contrast our findings with prior literature. The citations have been updated.

Conclusions

The conclusion is not specific and not aligned with the findings. Please first write the word in full before you abbreviated it such as LMICs. You recommend to LMICS, but I couldn’t see any evidence in the method sections including sampling technique, that support for generalizability of the findings for LMICs. The conclusion is beyond the study’s scope and not specific. Please revise it. Thank you. We have revised the conclusions to reflect our findings.

REVIEWER 3 

This is a comprehensive study looking at antibiotic consumption patterns in a LMIC. It is well-written except for a few grammatical errors (Introduction - line 3, page 5 - line 10, page 20 - lines 1 and 6). Thank you. The paper has been revised extensively and we have corrected the errors stated.

Data however is far from complete since many potential sources have not been able to provide usable data (ie only 12/78 private importers have submitted usable data). Agree and we have explained this in our limitations.

Although there were 78 importers registered with NMRA, the exact number of Companies imported antibacterials in 2017 was not available: Rough estimate from NMRA records was 40. Of them, we have got data from 12. Moreover, of the 5 leading importers, we have got data from 4. In addition, the private sector distribution data from SPC could have captured the data for the Companies who have not submitted data to us

We have mentioned in the results section, paragraph 1

A major factor that affects antibiotic consumption in the public hospitals in Sri Lanka is the highly limited formulary that is available for prescribers, compared to the private sector. This fact has not been taken in to account in the Discussion. Being able to 'locally purchase' antibiotics in the public sector does not equate to the choice of antibiotics available in the private sector. Thank you for this suggestion. We have incorporated it to the discussion as below:

A major factor that affects antibiotic consumption in the public hospitals in Sri Lanka is the highly limited formulary available to prescribers. The formulary is based on the WHO’s and the country’s EMLs and the medicines are made available strictly according to it to the public sector institutions. Even though prescribers have the facility of “local purchase” of medicines, this does not give them the same availability of medicines as those practicing in the private sector as the patients who seek care from these institutions are largely those with limited financial means.

No information on antibiotic resistance patterns in the country is provided. Therefore, the usefulness and applicability of the prescription patterns is not explored in detail. Thank you for this suggestion. We have added information on AMR patterns in Sri Lanka both in the Background and results and discussion sections.

Additional changes done by us 

a) Surveillance was changed to survey in all places

---

## [Decision Letter · Decision Letter 1]

6 Jul 2021

PONE-D-21-11873R1

A National Survey of antibacterial consumption in Sri Lanka

PLOS ONE

Dear,

Thank you for submitting your manuscript to PLOS ONE. After careful consideration, we feel that it has merit but does not fully meet PLOS ONE’s publication criteria as it currently stands. Therefore, we invite you to submit a revised version of the manuscript that addresses the points raised during the review process.

Please see reviewer comments and file attached.

We look forward to receiving your revised manuscript.

Kind regards,

Muhammad Shahzad Aslam, Ph.D.,M.Phil., Pharm-D

Academic Editor

PLOS ONE

Journal Requirements:

Reviewers' comments:

Reviewer's Responses to Questions

**Comments to the Author**

1. If the authors have adequately addressed your comments raised in a previous round of review and you feel that this manuscript is now acceptable for publication, you may indicate that here to bypass the “Comments to the Author” section, enter your conflict of interest statement in the “Confidential to Editor” section, and submit your "Accept" recommendation.

Reviewer #4: All comments have been addressed

Reviewer #5: (No Response)

2. Is the manuscript technically sound, and do the data support the conclusions?

Reviewer #4: Partly

Reviewer #5: Yes

3. Has the statistical analysis been performed appropriately and rigorously? 

Reviewer #4: Yes

Reviewer #5: Yes

4. Have the authors made all data underlying the findings in their manuscript fully available?

Reviewer #4: Yes

Reviewer #5: Yes

5. Is the manuscript presented in an intelligible fashion and written in standard English?

Reviewer #4: Yes

Reviewer #5: Yes

6. Review Comments to the Author

Reviewer #4: section 2.

The manuscript is technically sound, and the information presented were rigorously analyzed.

These are the points that I want the authors to better clarify and that is why I feel there is a little information that is lacking. I also want to recall that the authors specified the scarcity of information related to the topic they surveyed. So, I think the comment I will make here are somehow important for readers to get the most of what was shown, but if it not possible for the authors to fully answered my concerns, then I am not sure it will change the great work that was done:

PAGE 13: it is acknowledged that the public sector provides most inpatient care compared to the private sector, but it is not well explained what is the link/association between inpatient care and high ABC!

Did you ascertain the proportion of outpatient prescription of drugs with that of inpatient? If not, the argument on inpatient needs to be supported by specific references.

The 74.5% of imports to the private sector is in favor of the idea “that the private sector use more drugs for outpatient care than inpatient care”.

What could justify a higher use of macrolides, quinolones and other beta-lactams in the private sector compare to the public?

As all these aspects will have policy implications, it is important to answer (or at least try) these points.

What could also explain the disproportionately higher use of broad spectrum antibacterial in the private sector than in the public?

Page 20 and 21.

[REASONS FOR GREATER USE BRAOD SPECTRUM ANTIBACTERIAL IN THE PRIVATE SECTOR: lack of regulatory oversight, greater accessibility to antibacterial and improved financial status of patients.

REASONS FOR GREATER USE BRAOD SPECTRUM ANTIBACTERIAL IN THE PUBLIC SECTOR: inappropriate prescriptions of antibacterial, incorrect physician perception of the need for antibacterial, high patient volume, fear of bacterial super infection and the limited formulary available to prescriber.] It is difficult to understand how each of these factors affect the use in a different manner in the two different studied setting.

Finally, is it possible to know if in both private or public hospitals, there are some guidelines on antibiotic usage that are put in place to guide physicians or pharmacists in healthcare facilities? The idea is to know if there are some protocols to follow before or during prescription.

Reviewer #5: This is an appreciable work done on a very important reasearch area that acts as a compus for the AMR surveillance strategies . I have made some queries and comments in the annotated file and would appreciate that these are attended to ensure more clarity and for ease in understanding the data processing and interpretation for the complex area of AMC. The readers must be able to understand the data and study design in the context of the country where study is conducted. It is appreciable tha the authors have addressed the comments of previous eviewers diligently

7. PLOS authors have the option to publish the peer review history of their article (what does this mean?). If published, this will include your full peer review and any attached files.

Reviewer #4: No

Reviewer #5: No

---

## [Author Response · Author response to Decision Letter 1]

28 Jul 2021

Journal Requirements: 

Please review your reference list to ensure that it is complete and correct. If you have cited papers that have been retracted, please include the rationale for doing so in the manuscript text, or remove these references and replace them with relevant current references. Any changes to the reference list should be mentioned in the rebuttal letter that accompanies your revised manuscript. If you need to cite a retracted article, indicate the article’s retracted status in the References list and also include a citation and full reference for the retraction notice

 Thank you 

We have reviewed the reference list. It was complete and correct, none of the articles we have cited have been retracted. Unfortunately, WHO has migrated some of their documents and the links had changed after 18th May 2021 which was the time of our initial submission. We have updated the links as of 9th July 2021. 

Some of the references from the WHO have compatibility issues with some web browsers and may not be seen with browsers such as Safari.

Reviewer 3 PONE-D-21-11873 

These comments are for the older version PONE-D-21-11873 and have been addressed in our submission PONE-D-21-11873R1

Introduction – highlighted words to be deleted This has been addressed in the resubmission PONE-D-21-11873R1

Add another limitation: the fact that you only calculated the doses assuming that only adults used the antibiotics. This has been addressed in the resubmission PONE-D-21-11873R1 (Page 22)

Now, we have added under limitations also (Page 23)

Reviewer 4 

PAGE 13: it is acknowledged that the public sector provides most inpatient care compared to the private sector, but it is not well explained what is the link/association between inpatient care and high ABC!

 Thank you, Similar to many resource limited countries, in Sri Lanka too, use (and overuse) of ABAs is high in inpatient care. This is because majority of infectious diseases which need antibiotics (septicaemia, meningitis, complicated dengue, pneumonia, etc.) are admitted to hospitals, whereas only patients with mild infections (bacterial pharyngitis, bacterial otitis media, uncomplicated pneumonias and UTIs etc.) are treated as outpatient. In addition, there are limitations in doing bacteriological investigations in outpatient settings. Hence patients that need bacteriological investigations are admitted. Therefore, it is natural to expect high volume of ABC in public sector which handles majority of inpatient care in Sri Lanka. However, this was not the case in our results which showed a higher ABC in the private sector.

Therefore, we have modified the statement to reflect our observation more clearly

Previous statement 

Despite most of the inpatient care being provided by the public sector health care institutions, the ABC in the private sector (Table 1; data sources 1, 4 and 7) accounted for 246.76 million DDDs compared to 97.96 million DDDs distributed to the public sector by the MSD (data source 2). 

Edited statement in the current submission (Page 12)

In Sri Lanka, most of the inpatient care is provided by the public sector while the outpatient care is shared between the public and private sectors [15]. Despite bulk of patient care being provided by the public sector, the ABC in the private sector (Table 1; data sources 1, 4 and 7) was 246.76 million DDDs compared to 97.96 million DDDs distributed to the public sector by the MSD (data source 2).

Did you ascertain the proportion of outpatient prescription of drugs with that of inpatient? If not, the argument on inpatient needs to be supported by specific references. As we have used procurement data, we did not ascertain the proportion of outpatient and inpatient use. However, in our experience about healthcare structure in Sri Lanka, major infectious diseases (confirmed and suspected) are managed as inpatients in public sector in Sri Lanka. Despite this fact, public sector consumption was only 97.96 million DDDs. Public sector’s greater contribution to inpatient care is supported by reference [15]

The 74.5% of imports to the private sector is in favor of the idea “that the private sector use more drugs for outpatient care than inpatient care”. Thank you and we agree. The following has been added to explain the situation in Sri Lanka. As there are no published studies in this area we have stated our experiences and used the words ‘could be”

Added paragraph (Page 12-13)

The higher ABC of private sector could be due to the significant proportion of outpatient care that is provided by the private sector which includes the private hospitals and independent family physicians in the community. The antibiotics prescribed by these sectors are obtained from the retail pharmacies in the private hospitals and in the community and will contribute towards the increased ABC of the private sector.

What could justify a higher use of macrolides, quinolones and other beta-lactams in the private sector compare to the public?

As all these aspects will have policy implications, it is important to answer (or at least try) these points.

 Thank you for the observation. While nothing can justify this high use of broad spectrum and Watch antibiotics by the private sector, we have tried to explain this in the discussion (page 21)as given below:

A significant finding from our survey is the disproportionately higher use of broad-spectrum antibiotics in the private sector when compared to the ABC of the public sector. The disparity is difficult to explain, especially as prescribers in both these sectors being largely the same. In addition, the National Antibiotic Guidelines for empirical treatment of infections [31] have been widely disseminated to the prescribers and they are expected to adhere to these irrespective of the place of practice. A major factor that affects antibacterial consumption in the public hospitals in Sri Lanka is the highly limited hospital formulary available to prescribers. The formulary is based on the country’s EMLs and the medicines are made available strictly according to it to the public sector institutions. Even though prescribers have the facility to “local purchase” medicines, this does not give them the same availability of medicines as those practicing in the private sector as the patients who seek care from these institutions are largely from lower income segments of the society. The freedom to prescribe any medicine that is available in the country and improved financial status of the patients who seek treatment from the private sector could contribute to the greater use of broad-spectrum and newer antibiotics in the private sector. 

What could also explain the disproportionately higher use of broad spectrum antibacterial in the private sector than in the public?

 Thank you for the comment, we have added the following in page 19 and the possible reasons are discussed in pages 21 in detail (see above comment)

While both sectors have consumed large amounts of oral and parenteral antibacterials in the Watch category, this is much more in the private sector. Notable is the disparity in the consumption of ceftriaxone, cefuroxime and meropenem. This is a concern as it would contribute significantly towards the spread of antibiotic resistant organisms in the country

Page 20 and 21.

[REASONS FOR GREATER USE BRAOD SPECTRUM ANTIBACTERIAL IN THE PRIVATE SECTOR: lack of regulatory oversight, greater accessibility to antibacterial and improved financial status of patients.

REASONS FOR GREATER USE BRAOD SPECTRUM ANTIBACTERIAL IN THE PUBLIC SECTOR: inappropriate prescriptions of antibacterial, incorrect physician perception of the need for antibacterial, high patient volume, fear of bacterial super infection and the limited formulary available to prescriber.] It is difficult to understand how each of these factors affect the use in a different manner in the two different studied setting.

 Thank you. We have explained the reasons in the discussion (Pages 19-21.) “lack of regulatory oversight” has been removed as it does not give the correct picture.

Paragraph in the current version: 

A significant finding from our survey is the disproportionately higher use of broad-spectrum antibiotics in the private sector when compared to the ABC of the public sector. The disparity is difficult to explain, especially as prescribers in both these sectors being largely the same. In addition, the National Antibiotic Guidelines for empirical treatment of infections [31] have been widely disseminated to the prescribers and they are expected to adhere to these irrespective of the place of practice. A major factor that affects antibacterial consumption in the public hospitals in Sri Lanka is the highly limited hospital formulary available to prescribers. The formulary is based on the country’s EMLs and the medicines are made available strictly according to it to the public sector institutions. Even though prescribers have the facility to “local purchase” medicines, this does not give them the same availability of medicines as those practicing in the private sector as the patients who seek care from these institutions are largely from lower income segments of the society. 

The freedom to prescribe any medicine that is available in the country and improved financial status of the patients who seek treatment from the private sector could contribute to the greater use of broad-spectrum and newer antibiotics in this setting.

This statement is based on two studies done to assess antibiotic prescription in the outpatient setting of the public sector in Sri Lanka.

We have modified the sentences as given below for clarity: (Page 21-22)

However, inappropriate prescriptions of antibacterials are seen in the outpatient management of respiratory infections in the public sector hospitals too [32,33]. Patients demand for antibiotics as a “quick fix” for infections, incorrect physician perception of the need for antibacterials, fear of bacterial superinfection in viral diseases and the high patient volume in the outpatient settings which limits time for assessment have led to a greater prescription of broad-spectrum antibiotics [ 33]. 

Finally, is it possible to know if in both private and public hospitals, there are some guidelines on antibiotic usage that are put in place to guide physicians or pharmacists in healthcare facilities? The idea is to know if there are some protocols to follow before or during prescription. Thank you and yes there are national guidelines for empirical and prophylactic use of antimicrobials [31]. We have added this. 

Reviewer #5 - PONE-D-21-11873R1

Reviewer #5: This is an appreciable work done on a very important research area that acts as a compus for the AMR surveillance strategies. I have made some queries and comments in the annotated file and would appreciate that these are attended to ensure more clarity and for ease in understanding the data processing and interpretation for the complex area of AMC. The readers must be able to understand the data and study design in the context of the country where study is conducted. It is appreciable that the authors have addressed the comments of previous reviewers diligently Thank you

Page 10 Para 1

Describe if any list was provided to the respondents or they were left to decide by themselves for medicines with antibacterial effect. It is presumed that the data was invited for all antibacterials irrespective of their EML listing status/.Please clarify Thank you, Yes, data were requested irrespective of their EML/Formulary status. We provided the ATC code {antibacterials for systemic use (J01)} for those who had the data in ATC format (e.g. MSD, SPC). For others, we provided a list containing the names of antibacterials categorized under TC code antibacterials for systemic use (J01)

We have added the following sentence in page 8

Details of all antibacterials were requested irrespective of their EML or Sri Lankan Formulary listing status. We provided the ATC code (J01) for those who had the data in the ATC format and a list containing the names of antibacterials categorized under J01 for those who did not have the data in the ATC format.

Page 10 - Methodology

methodology states that data i calculated in DDDs using adult dosages. Were consumption data adjusted for syrups excluding them or keeping a percentage. What is the Sri Lankas population proportion for paediatric and geriatric patients Thank you for the comment 

In Sri Lanka, liquid preparations are not used in geriatric population as they are expensive. Since the dose required for elderly is higher, the requirement of liquid preparations could be many times more than that is required for children. Neither the country nor the great majority of patients can afford this. Even for children, liquid preparations are not uniformly available (Indian J Pediatr 2021. Feb;88(2):178-179.doi: 10.1007/s12098-020-03409-6. Epub 2020 Jun 20. 

Percentage of children (0-5 years) in Sri Lanka is about 8.5%

Using the DDDs for adults when there is paediatric consumption also is an inherent limitation of drug utilization surveys. We have noticed this limitation in other published articles as well. Hence, we have not done any adjustment. We have stated this as a limitation (Page 22, 23) 

Mention, in Limitation the concerns related to paediatric patients Thank you, we had done this in PONE-D-21-11873R1 under limitations in page 22 and this is what we have said:

“However, the DDD may not reflect the prescribed daily dose (PDD) for individual patients and cannot be used to measure consumption in paediatric wards since the measure is based on adult dosing [36]. It also does not accurately measure antibacterial consumption in cases of renal or hepatic dysfunction, often underestimating the actual antibacterial usage [36].”

Please explain how you have identified duplication of data while the same consumption of supply might be mentioned by manufacturer/importer and the distributor. Figure 1 shows the supply chain of medicines in Sri Lanka and it shows that there is minimal chance of duplication of data. MSD is the sole supplier of medicines to the public sector. Limitation was “non-responders” and “submission of incomplete data that could not be analysed” mainly the private importers. We have mentioned this in the Results (page 20) and limitations (page 22/23)

Mention the source (This is given near Aware Classification) We have modified this as follows in page ….

AWaRe classification 2019 [9] Page 4

Which AWaRe classification if from WHO was used . If it was AWaRe 207 what strategy was adopted for class assignment to medicine shared in access and watch group both As cited, we have used the 2019 AWaRe classification

Ethical Consideration Page 10

at any place has it been mentioned that the institutional and personnel identity contributing to the data will be kept anonymous and coded names or ID will be used in data acquisition and processing

 Thank you for the comment. We have not mentioned this. This national survey was done as a service requirement and the identity of the participating institutions was essential to identify the gaps. As such the participating institutions were not deidentified and they have consented to submit data with that understanding. This is the reason for us not to publicly share the raw data as requested by the Journal.

The personnel providing the data have done it in their official capacity and have been identified only as the Institution.

Results and discussion

Use lower case for companies please Thank you, we have corrected these typos.

Though dosage form data had been collected but still no discussion and results were shared to help interpret the results and form comparison among the two sectors. 

 Thank you. We had discussed all our findings. Tables 5 & 6 compares the top 10 oral and parenteral ABC in the two sectors. The following has been added (page 19)

While both sectors have consumed large amounts of oral and parenteral antibacterials in the Watch category, this is much more in the private sector. Notable is the disparity in the consumption of ceftriaxone, cefuroxime and meropenem. This is a concern as it would contribute significantly towards the spread of antibiotic resistant organisms in the country.

Again the point that if the data is not adjustable to accommodate the paediatric doses (and only used adult DDDs) then why syrup and suspension formulations remained included the consumption data. The proportion could have been easily shared here in data to access the margin of error created, even if pooled data (Oral liquid forms inclusive data) was used for the discussion of the overall data.

 Thank you 

As indicated earlier, liquid preparations are not extensively used due to cost difference

This is the proportion of liquid preparations in different data sources (in million DDDs). DDDs for liquid preparations were also calculated using adult DDD

 Total Liquid 

Distributed by SPC to private sector 163.04 19.75 

Distributed by MSD to public sector 97.96 1.56

Imported by private sector 61.18 None 

Distributed by SPMC to both to public and private sector 59.30 3.2 

Manufactured by SPMC 56.08 None 

Distributed by SPC to MSD 31.43 1.15

Distributed by SPC to its retail pharmacies (Rajya Osusala) 24.41 1.13 

Only a negligible amount has been contributed by the liquid preparations 

It is understandable that the data acquisition in this area is a difficult task. It will be helpful that the supply chain process or hierarchy be depicted pictorially and so that its clear to identify the niche of each respondent organization and see if the data has chance of being duplicated Thank you and we have added Figure 1: the antibiotic supply chain in Sri Lanka for clarity but would like to retain table 1 as it is important for the discussion.

At the moment its manufacturer, importer and distributor data under discussion. Countries vary in their distribution and supply chain processes. The Local purchase discussion do not fit into this context as this is presumably a retail and hospital level transaction Thank you and we agree. We have deleted the section on local purchase. The antibiotics obtained as local purchase are from retail pharmacies and this could have been captured elsewhere. 

Table 1

A graphic representation of the results as flow diagram would have been more helpful showing various sources and receiving sectors Thank you and we have added Figure 1: the antibiotic supply chain in Sri Lanka for clarity but would like to retain table 1 as it is important for the discussion.

References New reference 31 has been added and all references have been checked to ensure that they are current and are cited appropriately.

---

## [Decision Letter · Decision Letter 2]

18 Aug 2021

PONE-D-21-11873R2

A National Survey of antibacterial consumption in Sri Lanka

PLOS ONE

Dear,

Thank you for submitting your manuscript to PLOS ONE. After careful consideration, we feel that it has merit but does not fully meet PLOS ONE’s publication criteria as it currently stands. Therefore, we invite you to submit a revised version of the manuscript that addresses the points raised during the review process.

Please include the following items

1-Please provide detail limitation of study after discussion.

2-Please indicate generalizability of findings.

3-Please provide detail recommendation as separate heading for policy makers.

We look forward to receiving your revised manuscript.

Kind regards,

Muhammad Shahzad Aslam, Ph.D.,M.Phil., Pharm-D

Academic Editor

PLOS ONE

Journal Requirements:

Additional Editor Comments:

Please include the following items

1-Please provide detail limitation of study after discussion.

2-Please indicate generalizability of findings.

3-Please provide detail recommendation as separate heading for policy makers.

Reviewers' comments:

Reviewer's Responses to Questions

**Comments to the Author**

1. If the authors have adequately addressed your comments raised in a previous round of review and you feel that this manuscript is now acceptable for publication, you may indicate that here to bypass the “Comments to the Author” section, enter your conflict of interest statement in the “Confidential to Editor” section, and submit your "Accept" recommendation.

Reviewer #4: All comments have been addressed

Reviewer #6: All comments have been addressed

2. Is the manuscript technically sound, and do the data support the conclusions?

Reviewer #4: Yes

Reviewer #6: Yes

3. Has the statistical analysis been performed appropriately and rigorously? 

Reviewer #4: Yes

Reviewer #6: Yes

4. Have the authors made all data underlying the findings in their manuscript fully available?

Reviewer #4: Yes

Reviewer #6: Yes

5. Is the manuscript presented in an intelligible fashion and written in standard English?

Reviewer #4: Yes

Reviewer #6: Yes

6. Review Comments to the Author

Reviewer #4: (No Response)

Reviewer #6: (No Response)

7. PLOS authors have the option to publish the peer review history of their article (what does this mean?). If published, this will include your full peer review and any attached files.

Reviewer #4: No

Reviewer #6: No

---

## [Author Response · Author response to Decision Letter 2]

26 Aug 2021

Comment Response 

Please provide detail limitation of study after discussion 

Thank you 

During the last revision, we had already included the paragraph given below at the end of the discussion. We hope this is adequate. 

No changes have been made now 

“An important limitation of our study was the inability to capture all national data. This was largely due to incomplete and inadequate record keeping by the Customs and private importers. The inability of the SPMC to provide consumption data based on the sector to which it supplied added to the incomplete national consumption data. For meaningful interpretation of data, the total numbers of DDDs derived as consumption estimates should be adjusted for the population to which the data apply. Despite these limitations we have adjusted for the population (DID) to compare with similar studies as there is no separate DDD for children (21). Therefore, we had to use the DDDs for adults in the calculations although both adults and children would have consumed the antibacterials. “

Please indicate generalizability of findings Thank you 

We have added this statement at the end of discussion 

This is the first time an attempt has been made to document the national consumption of antibacterials in Sri Lanka. While there are some limitations and the actual consumption could be an under estimation, we are confident that the pattern of antibacterial use documented in this paper is unlikely to change even if we have all the whole data on antibacterial consumption. The generalizability of our findings would depend on the systems in place to regulate and survey antibacterial consumption. The paper highlights the need for better regulation of antibacterial consumption and the need for robust surveillance systems. The latter could be both labour and resource intense to LMICs like Sri Lanka.

Please provide detail recommendation as separate heading for policy makers Thank you 

We have added the following paragraphs at the end of the body of the manuscript. A new reference is added (38) 

Recommendations for policy makers

We strongly recommend the establishment of robust and sustainable surveillance systems to periodically survey and monitor antibacterial consumption. A central body to coordinate the activities of antibacterial consumption is crucial. Surveillance systems should be developed, and adequate funding and resources to collect and analyze data should be made available. All data should be coded at the point of entry using the ATC classification which would help in analysis of data and to compare the consumption trends with other countries.

The data on ABC should be linked with that of AMR to identify trends of antibacterial use and changes in antibacterial sensitivity patterns. The ABC should be reviewed annually to identify trends of use and to regularize antibacterial consumption. Based on the surveillance data, national policies and guidelines for antibacterial use should be developed and measures should be in place to ensure that they are adhered to. Linking up with the WHO programme that has been introduced for LMICs [38] is important to compare the country’s activities with others.

---

## [Editor Report · Decision Letter 3]

1 Sep 2021

A National Survey of antibacterial consumption in Sri Lanka

PONE-D-21-11873R3

Dear Dr. Sri Ranganathan,

We’re pleased to inform you that your manuscript has been judged scientifically suitable for publication and will be formally accepted for publication once it meets all outstanding technical requirements.

Kind regards,

Muhammad Shahzad Aslam, Ph.D.,M.Phil., Pharm-D

Academic Editor

PLOS ONE
---

## [Editor Report · Acceptance letter]

6 Sep 2021

PONE-D-21-11873R3 

A National survey of antibacterial consumption in Sri Lanka 

Dear Dr. Sri Ranganathan:

I'm pleased to inform you that your manuscript has been deemed suitable for publication in PLOS ONE. Congratulations! Your manuscript is now with our production department. 

Kind regards, 

on behalf of

Dr. Muhammad Shahzad Aslam 

Academic Editor

PLOS ONE